# SCALABLE PRINCIPAL–AGENT CONTRACT DESIGN VIA GRADIENT-BASED OPTIMIZATION

## ABSTRACT

We study a bilevel *max–max* optimization framework for principal–agent contract design, in which a principal chooses incentives to maximize utility while anticipating the agent's best response. This problem, central to moral hazard and contract theory, underlies applications ranging from market design to delegated portfolio management, hedge fund fee structures, and executive compensation. While linear–quadratic models such as Holmström–Milgrom admit closed-form solutions, realistic environments with nonlinear utilities, stochastic dynamics, or high-dimensional actions generally do not.

We introduce a generic algorithmic framework that removes this reliance on closed forms. Our method adapts modern machine learning techniques for bilevel optimization—using implicit differentiation with conjugate gradients (CG)—to compute hypergradients efficiently through Hessian–vector products, without ever forming or inverting Hessians. In benchmark CARA–Normal (Constant Absolute Risk Aversion with Gaussian distribution of uncertainty) environments, the approach recovers known analytical optima and converges reliably from random initialization. More broadly, because it is matrix-free, variance-reduced, and problem-agnostic, the framework extends naturally to complex nonlinear contracts where closed-form solutions are unavailable, such as sigmoidal wage schedules (logistic pay), relative-performance/tournament compensation with common shocks, multi-task contracts with vector actions and heterogeneous noise, and CARA–Poisson count models with $\mathbb{E}[X|a] = e^a$. This provides a new computational tool for contract design, enabling systematic study of models that have remained analytically intractable.

## 1 INTRODUCTION

The design of incentive mechanisms is a central problem in economics, finance, and operations research (Bolton and Dewatripont, 2005; Lazear and Gibbs, 2014; Milgrom and Roberts, 1992; Jensen and Meckling, 1976; Cachon, 2003; Laffont and Martimort, 2002). In many settings, a *principal* (e.g., an employer, regulator, firm, or portfolio manager) seeks to influence the actions of an *agent* (e.g., an employee, contractor, or service provider) whose choices directly affect the principal's payoff (Grossman and Hart, 1983). A key challenge is that the principal cannot directly dictate the agent's decision; instead, they must offer a *contract* specifying how the agent will be compensated based on observable outcomes (Salanié, 2017; Holmström, 1979; Holmström and Milgrom, 1987). The agent, upon observing the contract, chooses an action that maximizes their own utility, which may diverge from that of the principal.

This interaction leads naturally to a *bilevel optimization* problem in which both levels are maximization problems: the principal optimizes contract parameters in the outer problem, anticipating the agent's best-response in the inner problem (Colson et al., 2007). Such models appear throughout the literature on *moral hazard* (Holmström, 1979; Grossman and Hart, 1983; Holmström and Milgrom, 1987), mechanism design (Myerson, 1981), and industrial organization.

A canonical example is the *linear–quadratic principal–agent model* of Holmström and Milgrom (Holmström and Milgrom, 1987), in which the principal offers a linear contract $t = (s, b)$, consisting of a fixed payment $s$ and a performance-based incentive $b$. The agent chooses an effort level $a$ that is costly to exert, and output is noisy. Under quadratic cost of effort and mean–variance preferences, this model admits a closed-form solution for $(s^\star, b^\star, a^\star)$ (Holmström and Milgrom,

1987). While this structure is analytically tractable, real-world contracts often involve nonlinear utilities, richer stochastic dynamics, and high-dimensional actions for which closed-form solutions do not exist (Sannikov, 2008).

When $u_1$ and $u_2$ are differentiable, a natural computational strategy is to use gradient-based optimization (Domke, 2012). However, in bilevel settings the outer objective $u_1(a^\star(t), t)$ depends on the contract parameters $t$ both directly and indirectly through the agent's optimal response $a^\star(t)$. Differentiating through the inner maximization requires computing *hypergradients* that involve inverting the Hessian of $u_2$ with respect to the agent's action, a computational bottleneck in high dimensions (Gould et al., 2016).

## 1.1 RELATED WORK

**Principal–agent theory and contract design.** Classic insight into incentive contracts arises from hidden–action models of moral hazard (Holmström, 1979; Grossman and Hart, 1983; Holmström and Milgrom, 1987) and the broader mechanism–design tradition (Myerson, 1981). These foundations have been synthesized in comprehensive works spanning labor economics, industrial organization, management, and finance (Bolton and Dewatripont, 2005; Laffont and Martimort, 2002; Salanié, 2017; Lazear and Gibbs, 2014; Milgrom and Roberts, 1992; Bard, 2008; Ho et al., 2014). In the canonical CARA–Normal, linear–contract framework, the Holmström–Milgrom model offers closed-form solutions under analytical tractability (Holmström and Milgrom, 1987). Extensions have preserved tractability while adding features such as multitask incentives (Holmström and Milgrom, 1991), imperfect or multiple performance measures (Baker, 1992; Laffont and Martimort, 2002), and insurance with prevention trade-offs (Ehrlich and Becker, 1972; Shavell, 1979). While these classical models frequently admit closed-form solutions, they do so under restrictive assumptions—quadratic costs, Gaussian uncertainty, and linear contracts—that constrain their applicability in nonlinear or high-dimensional settings.

**Algorithmic contract theory.** Since closed-form solutions are often infeasible, the emerging field of algorithmic contract theory (Duetting et al., 2024; Dütting and Talgam-Cohen, 2019) develops computational tools for a broader range of principal–agent settings. Much of this literature focuses on discrete action spaces. One strand analyzes the performance of linear contracts, establishing approximation and robustness guarantees relative to the optimal benchmark (Dütting et al., 2019). Another examines combinatorial and multi-agent contracts, where the exponential size of the action space motivates approximation algorithms, query-complexity analyses, and hardness results (Dütting et al., 2021; Ezra et al., 2024; Cacciamani et al., 2024). Typed-agent models motivate menu-based contracts, yielding welfare and approximation guarantees under single-dimensional and thin-tail assumptions (Alon et al., 2021; 2023).

Beyond these discrete formulations, more recent work leverages learning and continuous optimization. Data-driven approaches provide regret bounds and statistical guarantees, while neural methods approximate the principal's utility and optimize over contract spaces. For example, Wang et al. (2023) learn piecewise-affine surrogates of payoffs and use gradient-based inference to recover contracts, while the differentiable economics framework (Dütting et al., 2024) parameterizes mechanisms with neural networks and trains them end-to-end via stochastic gradient descent. Other directions explore contracts for strategic effort in machine learning (Babichenko et al., 2024), ambiguous or incomplete contracts (Duetting et al., 2023). Collectively, these results chart a broad algorithmic landscape that complements classical theory by emphasizing approximation, learnability, and computational tractability.

**Bilevel optimization and differentiable methods.** Abstractly, principal–agent models can be cast as bilevel *max–max* programs where the principal's optimization anticipates the agent's best response. Bilevel optimization has a long tradition in operations research (Colson et al., 2007) and has become central in machine learning—powering hyperparameter tuning, meta-learning, data reweighting, and differentiable programming (Lorraine et al., 2020; Rajeswaran et al., 2019; Ren et al., 2018). Early methods differentiated through the inner solver by unrolling optimization steps (Domke, 2012), which is memory-intensive and prone to truncation bias. More scalable approaches apply implicit differentiation to the inner optimality conditions, reducing the task to solving linear systems involving Hessians or Jacobians (Gould et al., 2016; Lorraine et al., 2020). Modern techniques combine automatic differentiation for Hessian–vector products with Krylov solvers like conjugate gradient

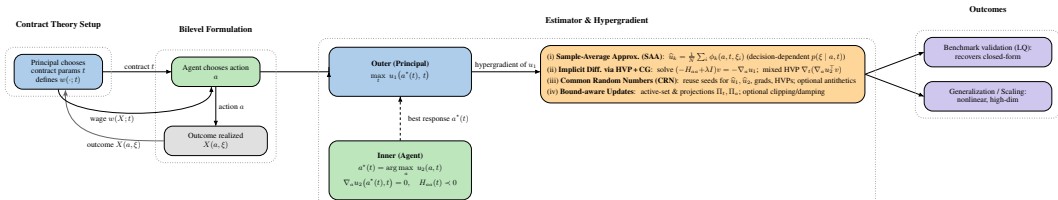

Figure 1: **Principal–agent bilevel with differentiable contracts.** *Left:* The principal chooses contract *parameters* $t \in \mathbb{R}^m$ that define a wage rule $w(x; t)$. The agent, after observing $t$, selects an action $a$. A stochastic outcome $X$ is realized according to $X = X(a, \xi)$ with $\xi \sim P(\cdot \mid a, t)$, and the wage $w(X; t)$ is paid. *Middle:* The principal solves $\max_t u_1\big(a^\star(t), t\big)$ anticipating the agent's best response $a^\star(t) \in \arg\max_a u_2(a, t)$, with first-order condition $\nabla_a u_2(a^\star(t), t) = 0$ and local curvature $H_{aa} \prec 0$. *Right:* Hypergradients $\nabla_t u_1(a^\star(t), t)$ are computed via sample-average approximation, implicit differentiation using Hessian–vector products with conjugate gradients (matrix-free%), common-random-number variance reduction, and bound-aware updates.

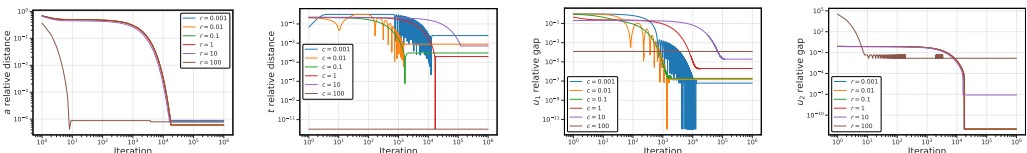

Figure 2: **Results for the Holmström–Milgrom linear–quadratic model with varying $r$.** Fixed parameters: $c = 1.0$, $\sigma = 0.1$; varied parameter: $r \in \{10^i \mid i = -3, \ldots, 2\}$.

(CG) (Hestenes and Stiefel, 1952; Barrett et al., 1994), enabling scalable, matrix-free hypergradient computation (Maclaurin et al., 2015; Rajeswaran et al., 2019).

### 1.2 CONTRIBUTIONS

In contrast to approaches that learn neural surrogates of the principal's utility and then optimize them via gradient-based inference (Wang et al., 2023), or that parameterize mechanisms with neural networks and train them end-to-end using stochastic gradient descent (Dütting et al., 2024), our method directly addresses the bilevel *max–max* structure of hidden-action problems. Whereas surrogate-based methods approximate the objective and depend on network training to suggest good contracts, our framework computes exact hypergradients through the agent's best-response conditions using implicit differentiation. This distinction is crucial: it enables a general-purpose solver for principal–agent problems that (i) recovers classical closed-form solutions with minimal approximation error, and (ii) scales to nonlinear and high-dimensional settings without discretization.

## 2 PROBLEM SETUP

We study the problem of *contract design* in a principal–agent relationship (Grossman and Hart, 1983; Holmström, 1979). In this framework, a *principal* (e.g., an employer, firm, or regulator) seeks to design an incentive scheme that shapes the behavior of an *agent* (e.g., an employee, contractor, or service provider).

The interaction has four defining features: **(a)** the principal cannot directly dictate the agent's action $a$ (e.g., effort); **(b)** instead, the principal offers a *contract* $t$ that specifies a fixed payment together with performance-based incentives; **(c)** the agent, after observing $t$, selects $a$ to maximize their own utility $u_2(a, t)$, which includes both rewards and *penalties* (e.g., costs of effort); and **(d)** the principal's utility $u_1(a, t)$ depends on $t$ and on the induced action $a$.

Formally, $t \in \mathbb{R}^m$ denotes the contract parameters and $a \in \mathbb{R}^n$ the agent's action. Outcomes are stochastic: we write $\xi \in \Xi$ for exogenous uncertainty with law $\xi \sim P(\cdot | a, t)$ that may depend on

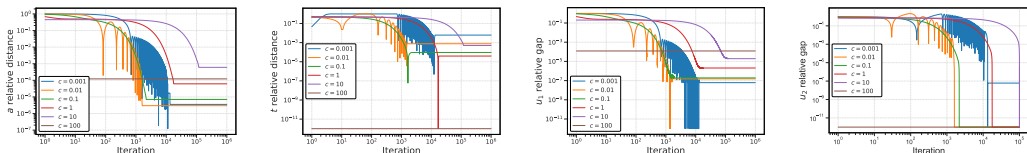

Figure 3: **Results for the Holmström–Milgrom linear–quadratic model with varying $c$.** Fixed parameters: $r = 1.0$, $\sigma = 0.1$; varied parameter: $c \in \{10^i \mid i = -3, \ldots, 2\}$.

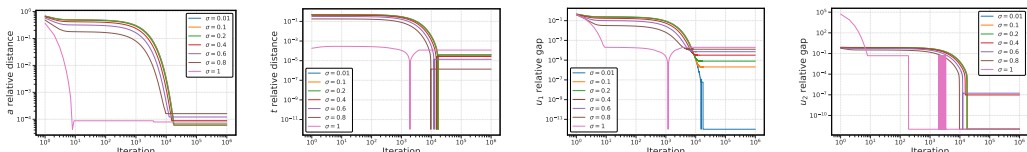

Figure 4: **Results for the Holmström–Milgrom linear–quadratic model with varying $\sigma$.** Fixed parameters: $r = 1.0$, $c = 1.0$; varied parameter: $\sigma \in \{0.01, 0.1, 0.2, 0.4, 0.6, 0.8, 1\}$.

$(a, t)$. The expected utilities are

$$u_2(a,t) \;=\; \mathbb{E}_{\xi \sim P(\cdot \,|a,t)}\big[\phi_2(a,t,\xi)\big], \qquad u_1(a,t) \;=\; \mathbb{E}_{\xi \sim P(\cdot \,|a,t)}\big[\phi_1(a,t,\xi)\big], \tag{1}$$

where $C(a,t)$ represents the agent's penalty (e.g., effort cost, disutility, or risk). Anticipating the agent's response, the principal solves the bilevel max–max problem

$$\max_{t \in \mathbb{R}^m} \quad u_1\big(a^\star(t),\, t\big), \tag{2}$$

$$\text{s.t.} \qquad a^\star(t) \in \arg\max_{a \in \mathbb{R}^n} \; u_2(a,t). \tag{3}$$

This formulation imposes no specific wage rule, production technology, or participation constraint; the only structural requirement is that utilities are expressed as expectations under a possibly *decision-dependent* $P(\cdot \,|a,t)$.

In classical linear–quadratic models (e.g., Holmström–Milgrom), the forms of $u_1$ and $u_2$ yield closed-form expressions for $a^\star(t)$ and $t \mapsto u_1(a^\star(t), t)$. Outside these highly structured settings, however, realistic models involve nonlinearities, multidimensional actions, and additional constraints, precluding analytic solutions. We therefore seek a numerical method that (i) treats expectation-valued objectives faithfully, (ii) remains valid when $P(\cdot \,|a,t)$ depends on decisions, and (iii) scales to high-dimensional actions without forming or inverting Hessians. The central challenge is thus clear: ***How can we design efficient optimization algorithms to solve general bilevel max–max problems when no analytical solution is available?***

## 3 METHOD

We propose a *generic bilevel max–max optimizer* that integrates four key components: (i) sample-average approximation (SAA) for expectation-valued objectives; (ii) implicit differentiation of the inner argmax via matrix-free Hessian–vector products (HVPs) and conjugate gradients (CG); (iii) variance reduction using *common random numbers* (CRN); and (iv) bound-aware updates for the outer variables.

Throughout, we treat $a$ (agent variables) and $t$ (contract variables) as tensors of arbitrary shape. We assume that $u_2(\cdot, t)$ is differentiable and locally strictly concave in $a$ at the inner solution, while $u_1$ is differentiable in both arguments. Under these assumptions, the Hessian block $H_{aa}$ is negative definite at a local maximum of the inner problem, which ensures that $-H_{aa}$ is symmetric positive definite (SPD). This property is crucial, as it guarantees the stability of the CG solve used to compute implicit gradients.

### 3.1 Monte Carlo estimation with decision-dependent sampling

We approximate the expectations in equation 1 using a sample-average approximation (SAA). Given $(a, t)$ and i.i.d. samples $\xi_1, \dots, \xi_N \sim P(\cdot \,|\, a, t)$, the estimator is

$$\widehat{u}_k^{(N)}(a, t) \;=\; \frac{1}{N} \sum_{i=1}^N \phi_k(a, t, \xi_i), \qquad k \in \{1, 2\}. \tag{4}$$

When the sampling distribution depends on decisions, with density $p(\xi \,|\, a, t)$, the gradient of $u_k$ decomposes into explicit and score function terms. For $\theta \in \{a, t\}$ and under standard regularity conditions,

$$\nabla_\theta u_k(a, t) \;=\; \mathbb{E}\left[\nabla_\theta \phi_k(a, t, \xi) \;+\; \phi_k(a, t, \xi)\, \nabla_\theta \log p(\xi \,|\, a, t)\right]. \tag{5}$$

If a reparameterization $\xi = h(a, t, \varepsilon)$ with $\varepsilon \sim P_0$ exists (independent of learned parameters), we may instead compute gradients pathwise by differentiating $\phi_k(a, t, h(a, t, \varepsilon))$. In practice, we apply automatic differentiation directly to the SAA estimator equation 4, ensuring that analytic and Monte Carlo models share the same code path and that both score-function and pathwise gradients are handled seamlessly.

**Common Random Numbers (CRN) and consistent evaluation.** To reduce variance in hyper-gradients, we employ common random numbers (CRN). Each outer iteration generates a single randomness payload (mini-batch or latent seed) that is reused for all evaluations of $\widehat{u}_1, \widehat{u}_2$, their gradients, and all HVPs. Optionally, the seed is refreshed every $R$ steps, and antithetic augmentation ($z$ with $-z$) can be applied. For evaluation and reporting, we construct a single held-out batch that remains fixed throughout training. This *consistent evaluation* ensures that reported trajectories reflect parameter changes rather than sampling noise.

### 3.2 Implicit differentiation of the inner maximum

**Theoretical derivation.** Let $a^\star(t)$ denote a local maximizer of the agent's objective $a \mapsto u_2(a, t)$. At such a point, the first-order optimality condition and local curvature are

$$\nabla_a u_2\big(a^\star(t), t\big) \;=\; 0, \qquad H_{aa}(t) \;:=\; \nabla_{aa}^2 u_2(a^\star(t), t) \prec 0, \tag{6}$$

where the negative definiteness of $H_{aa}(t)$ reflects that $a^\star(t)$ is a strict local maximum.

The dependence of $a^\star(t)$ on the contract $t$ can be characterized using the implicit function theorem. Differentiating the optimality condition yields

$$\frac{\mathrm{d}a^\star(t)}{\mathrm{d}t} \;=\; -H_{aa}(t)^{-1}\, H_{ta}(t), \qquad H_{ta}(t) \;:=\; \nabla_{ta}^2 u_2(a^\star(t), t), \tag{7}$$

which provides the sensitivity of the best response to changes in the contract.

Substituting this expression into the chain rule gives the hypergradient of the outer objective

$$\nabla_t u_1(a^\star(t), t) \;=\; \nabla_t u_1(a^\star, t) \;-\; H_{ta}(t)\, H_{aa}(t)^{-1}\, \nabla_a u_1(a^\star, t). \tag{8}$$

Thus, computing the hypergradient requires solving a linear system involving $H_{aa}(t)^{-1}$.

Since $H_{aa}(t)$ is negative definite at the inner maximum, $-H_{aa}(t)$ is symmetric positive definite (SPD). We therefore solve for $v$ in the damped SPD system

$$(-H_{aa}(t) + \lambda I)\, v \;=\; -\nabla_a u_1(a^\star, t), \tag{9}$$

where a small damping parameter $\lambda \geq 0$ improves conditioning, with vanishing bias as $\lambda \to 0$.

Finally, the mixed Hessian term $H_{ta}(t)\, v$ is computed without explicitly forming Hessians, by applying the standard Hessian–vector product (HVP) identity

$$H_{ta}(t)\, v \;=\; \nabla_t(\nabla_a u_2\big(a^\star(t), t\big)^\top v). \tag{10}$$

This matrix-free formulation enables efficient computation at scale.

---

**Algorithm 1** Principal's Outer Loop

---

**Require:** Monte Carlo utilities $\widehat{u}_1, \widehat{u}_2$, initial $(t_0, a_0)$, outer steps $T_{\text{out}}$
1: $t \leftarrow t_0, \; a \leftarrow a_0$
2: **for** $k = 0$ **to** $T_{\text{out}} - 1$ **do**
3:      Build CRN payload (mini-batch or latent seed); reuse across all calls this step
4:      $\tilde{a} \leftarrow \text{INNERASCENT}(\widehat{u}_2, a, t)$
5:      $\text{hypergrad}_t \leftarrow \text{HYPERGRAD}(\widehat{u}_1, \widehat{u}_2, \tilde{a}, t)$
6:      $t \leftarrow \text{UPDATECONTRACT}(t, \text{hypergrad}_t)$
7:      $a \leftarrow \tilde{a}$                                         ▷ warm-start next inner loop
8:      Optionally log $\widehat{u}_1, \widehat{u}_2$ on held-out batch
9: **return** $t, a, \{\widehat{u}_1\}, \{\widehat{u}_2\}$, trace

---

**Algorithm 2** INNERASCENT: Agent Best Response with CRN

---

**Require:** Utility $\widehat{u}_2$, current $(a, t)$, inner steps $T_{\text{in}}$, step size $\eta_{\text{in}}$, tolerance $\varepsilon_{\text{in}}$, projection $\Pi_a$
1: **for** $\tau = 1$ **to** $T_{\text{in}}$ **do**
2:      $g_a \leftarrow \nabla_a \widehat{u}_2(a, t)$                                         ▷ CRN fixed
3:      **if** $\|g_a\| \leq \varepsilon_{\text{in}}$ **then break**
4:      $a \leftarrow \Pi_a\big(a + \eta_{\text{in}} g_a\big)$
5: **return** $\tilde{a} \leftarrow a$

---

### 3.3 PRACTICAL CONSIDERATIONS

**Approximating $a^\star(t)$.** In practice, the agent's exact best response $a^\star(t)$ cannot be computed. Instead, we approximate it by running a short gradient ascent on $u_2(\cdot, t)$, reusing the same CRN payload at every step to control variance. The ascent is terminated once the stationarity condition $\|\nabla_a u_2\|_2 \leq \varepsilon_{\text{in}}$ is satisfied, or after at most $T_{\text{in}}$ iterations. The resulting iterate $\tilde{a}(t)$ serves as an approximation of $a^\star(t)$ and also provides a natural warm start for the next outer iteration. Under mild regularity assumptions, the bias introduced by inexact solves vanishes as $\varepsilon_{\text{in}} \downarrow 0$.

**Hypergradient via implicit differentiation.** The hypergradient equation 8 is evaluated at $\tilde{a}(t)$ rather than $a^\star(t)$. To avoid explicitly inverting the inner Hessian $H_{aa}(t)$, we introduce an auxiliary vector $v$ defined as the solution of the damped SPD system

$$(-H_{aa}(\tilde{a}(t), t) + \lambda I)\, v \;=\; -\nabla_a u_1(\tilde{a}(t), t), \tag{11}$$

which is solved approximately using at most $T_{\text{cg}}$ iterations of conjugate gradient. The mixed Hessian term is then recovered in matrix-free form using the standard Hessian–vector product identity,

$$H_{ta}(t)\, v \;=\; \nabla_t \big(\nabla_a u_2(\tilde{a}(t), t)^\top v\big). \tag{12}$$

Combining these pieces yields the estimator

$$\text{hypergrad}_t \;=\; \nabla_t u_1(\tilde{a}(t), t) \;-\; \nabla_t \big(\nabla_a u_2(\tilde{a}(t), t)^\top v\big),$$

with optional norm clipping for numerical stability.

**SPD fix and damping.** At a local maximum of the inner problem, the curvature matrix $H_{aa}(t)$ is negative definite. Replacing it by $-H_{aa}(t)$ therefore yields a symmetric positive definite system, which is well suited for solution by CG. Adding a small damping term $\lambda I$ further improves conditioning, while the bias induced by damping vanishes in the limit $\lambda \to 0$.

**Variance reduction with CRN.** Within each outer iteration, all evaluations of $\widehat{u}_1, \widehat{u}_2$, their gradients, and every HVP reuse the same randomness payload (either a mini-batch or a latent seed). This reuse substantially stabilizes both the right-hand side of the CG system and the mixed HVP, yielding smoother optimization. In practice, seeds may be refreshed periodically (every $R$ steps), and antithetic augmentation (using $z$ and $-z$) can further reduce variance. For evaluation and reporting, a single fixed held-out batch is used throughout training to ensure comparability across iterations.

**Bounds and projections.** We support user-specified projections $\Pi_a, \Pi_t$; otherwise we apply box constraints $[\ell_b, u_b]$. Outer updates use an active-set scheme: coordinates strictly inside their

---

**Algorithm 3** HYPERGRAD: Implicit Differentiation via HVP + CG

---

**Require:** Utilities $\widehat{u}_1, \widehat{u}_2$, approximate best response $\tilde{a}$, contract $t$
**Require:** CG iterations $T_{\mathrm{cg}}$, damping $\lambda$, tolerance $\varepsilon_{\mathrm{cg}}$
 1: $g_a \leftarrow \nabla_a \widehat{u}_1(\tilde{a}, t)$                  ▷ outer grad wrt agent action
 2: $g_t \leftarrow \nabla_t \widehat{u}_1(\tilde{a}, t)$                        ▷ direct outer grad
 3: Define HVP operator $\mathcal{H}_{aa}[v] \leftarrow \nabla_a(\nabla_a \widehat{u}_2(\tilde{a}, t)^\top v)$
 4: $v \leftarrow \textsc{ConjugateGradient}((-\mathcal{H}_{aa} + \lambda I), -g_a, T_{\mathrm{cg}}, \varepsilon_{\mathrm{cg}})$
 5: $m \leftarrow \nabla_t(\nabla_a \widehat{u}_2(\tilde{a}, t)^\top v)$                 ▷ mixed Hessian term
 6: **return** hypergrad$_t \leftarrow g_t - m$

---

**Algorithm 4** UPDATECONTRACT: Bound-Aware Outer Update

---

**Require:** Current contract $t$, hypergradient hypergrad$_t$, step size $\eta_{\mathrm{out}}$, projection $\Pi_t$, active-set
  projector `active_set`
 1: $t \leftarrow t + \eta_{\mathrm{out}} \cdot$ hypergrad$_t$
 2: $t \leftarrow \texttt{active\_set}(t)$                 ▷ enforce box/bound constraints
 3: $t \leftarrow \Pi_t(t)$                       ▷ project into feasible domain
 4: **return** $t$

---

bounds update freely, while boundary coordinates are clamped. This preserves feasibility and avoids tangential drift along the constraint surface.

**Computational cost.** Let $n = \dim(a)$, $m = \dim(t)$, and $N$ the SAA batch size. Per outer iteration we compute $T_{\mathrm{in}}$ gradients w.r.t. $a$, $T_{\mathrm{cg}}$ HVPs with $-\nabla^2_{aa}\widehat{u}_2$, one mixed HVP for $\nabla_t(\nabla_a \widehat{u}_2^\top v)$, and the first-order gradients $\nabla_a \widehat{u}_1, \nabla_t \widehat{u}_1$. Under dense autodiff, a gradient/HVP w.r.t. the $a$-block scales linearly in $n$, and a gradient w.r.t. the $t$-block scales linearly in $m$. Hence the per-iteration work is $\mathcal{O}\big(N\left[(T_{\mathrm{in}} + T_{\mathrm{cg}} + 1)\, n \; + \; m\right]\big)$, with no Hessian formed or inverted. Memory is $\mathcal{O}(n)$ for CG vectors (plus model activations), and projection/active-set operations are $\mathcal{O}(m)$. Thus the method avoids the costs of explicit Hessians and scales linearly in $n$, $m$, and $N$.

## 4 EXPERIMENTS

**Experimental setup.** Our experiments evaluate whether implicit differentiation with conjugate gradient (CG) reliably recovers optimal contracts in principal–agent bilevel problems. We benchmark across two classes of environments:

- *Canonical CARA–Normal linear–contract models with closed-form solutions:* (**1**) the Holmström–Milgrom linear–quadratic benchmark (Holmström, 1979), (**2**) insurance with prevention (self-protection) (Ehrlich and Becker, 1972; Shavell, 1979), (**3**) imperfect performance measurement with a single noisy signal (Laffont and Martimort, 2002), (**4**) aggregation of two noisy signals (Laffont and Martimort, 2002; Baker, 1992), (**5**) separable multitask contracting (Holmström and Milgrom, 1991). These environments admit closed-form optima $(a^\star, t^\star)$, and (**6**) relative performance (peer benchmark), enabling direct error measurement.

- *Nonlinear signal environments without closed-form solutions:* Beyond the linear–quadratic CARA–Normal benchmarks, we also evaluate nonlinear environments motivated by the First-Order-Approach (FOA) literature, where explicit analytical solutions are unavailable. These include logistic and Poisson signal models and nonlinear wage utilities, which have been used as canonical worked examples in Jewitt's FOA analysis (Jewitt, 1988).

Full derivations of the CARA–Normal cases and details of the grid-search approximation for nonlinear signals are provided in Appendix B.1.

**Settings.** For each environment, we fix baseline parameters (e.g., cost $c$, noise $\sigma$) and vary one parameter of interest (e.g., risk aversion $r$) over a grid. Contract parameters $t$ and actions $a$ are initialized independently as $\mathcal{N}(0, 1)$. We solve the bilevel problem equation 2–equation 3 using Alg. 1. The agent's problem is solved by gradient ascent on $u_2$ with step size $\eta_{\mathrm{in}} = 5 \times 10^{-3}$ for at most $T_{\mathrm{in}} = 50$ iterations, terminating when $|\nabla_a u_2| \leq 10^{-4}$. The principal's parameters are updated

---

**Algorithm 5** CONJUGATEGRADIENT (CG Solver for SPD system)

---

**Require:** SPD operator $\mathcal{A}(\cdot)$, right-hand side $b$, max iters $T_{\mathrm{cg}}$, tolerance $\varepsilon_{\mathrm{cg}}$
1: $v \leftarrow 0$          ▷ initial guess (can also warm-start)
2: $r \leftarrow b - \mathcal{A}(v)$          ▷ residual
3: $p \leftarrow r, \quad \rho \leftarrow \langle r, r \rangle$
4: **for** $\tau = 1$ **to** $T_{\mathrm{cg}}$ **do**
5:     **if** $\sqrt{\rho} \leq \varepsilon_{\mathrm{cg}}$ **then break**
6:     $q \leftarrow \mathcal{A}(p)$
7:     $\alpha \leftarrow \rho / \langle p, q \rangle$
8:     $v \leftarrow v + \alpha p$
9:     $r \leftarrow r - \alpha q$
10:    $\rho_{\mathrm{new}} \leftarrow \langle r, r \rangle$
11:    $\beta \leftarrow \rho_{\mathrm{new}} / \rho$
12:    $p \leftarrow r + \beta p$
13:    $\rho \leftarrow \rho_{\mathrm{new}}$
14: **return** $v$

---

for $T_{\mathrm{out}} = 10^6$ (linear) and $T_{\mathrm{out}} = 10^5$ (nonlinear) steps with step size $\eta_{\mathrm{out}} = 10^{-3}$. Hypergradients are computed via CG with $T_{\mathrm{cg}} = 20$ iterations and damping $\lambda = 10^{-4}$. Each inner solve is warm-started from the previous $\tilde{a}(t)$. To control estimator variance, we use common random numbers (CRN): within each outer iteration we reuse a Sobol–QMC batch of size $n = 1024$ (with antithetic pairing), refreshing it every $K = 100$ iterations.

**Evaluation metrics.** For evaluation, we fix a Sobol–QMC batch of size 8,192 from the signal distribution and use it consistently to compute expected utilities. On this held-out batch, we report four primary metrics: the relative action and contract errors $\mathrm{err}_a = \frac{\|a - a^\star\|}{\|a^\star\| + \varepsilon}$, $\mathrm{err}_t = \frac{\|t - t^\star\|}{\|t^\star\| + \varepsilon}$ and the principal's normalized utility gap $\Delta u_1 = \frac{\left| u_1(a^\star, t^\star) - u_1(a, t) \right|}{\left| u_1(a^\star, t^\star) \right| + \varepsilon}$ and $\Delta u_2 = \frac{\left| u_2(a^\star(t), t) - u_2(a, t) \right|}{\left| u_2(a^\star(t), t) \right| + \varepsilon}$. Here, $\varepsilon$ is a small constant for numerical stability (we used $10^{-12}$ across all of the experiments).

**Ground-truth approximation.** For environments lacking closed-form solutions, we approximate the optimal contract by nested grid search. Specifically, we discretize the contract space $t$ over a box derived from the setting parameters, and for each candidate contract compute the agent's best response $a^\star(t)$ by maximizing $u_2(a, t)$ on a dense action grid. Utilities $u_1$ and $u_2$ are estimated by Monte Carlo expectation on a fixed Sobol–QMC batch of 8,192 samples from the relevant signal distribution (e.g., Logistic or Laplace). To reduce variance and ensure fair comparisons, the same random draws are reused across all grid points (common random numbers) and paired antithetically. The principal's payoff $u_1(a^\star(t), t)$ is then evaluated at each candidate, and the maximizing pair $(a^\star, t^\star)$ is taken as the approximate ground-truth solution. For example, in the logistic–signal case we discretize $(\lambda, \mu)$ over a rectangular box that scales with the signal scale $s$ (e.g. $\lambda \in [w_{\min}, w_{\min} + 8]$, $\mu \in [0, 8]$) using a $100 \times 100$ grid. For each $t = (\lambda, \mu)$, the best-response action is approximated on a 200-point grid spanning $[a_0 - 6s, a_0 + 6s]$. For per-setting details, see App. B.1.

For settings with multiple contract slopes, only the incentive coefficients are optimized by grid search, while fixed transfers $s$ are set post hoc to satisfy the participation constraint at equality. Wages are always floored at $w_{\min} > 0$ to maintain numerical stability under $\log$ or $\sqrt{\cdot}$ utilities. Together, these design choices yield stable and reproducible approximate optima that serve as a consistent reference for benchmarking our gradient-based solver.

## 4.1 RESULTS

Our experiments reveal a sharp distinction between linear and nonlinear environments. We first consider the family of linear CARA–Normal benchmarks. These settings admit closed-form solutions, allowing us to directly assess whether the learning dynamics recover both utilities and contract parameters. In every linear environment we studied, the results are unambiguous: the utility gaps converges very close to zero (e.g., 9, 12) and the distances for the learned contract parameters and the ground-truth parameters vanish (e.g., Figs. 8, 10, 14) as well as the distance between the learned

action and the corresponding optimal action. This demonstrates that the proposed bilevel solver with implicit differentiation is able to recover the analytic solution to high precision.

A key feature of these linear results is their robustness. Across sweeps in cost coefficients, levels of risk aversion, and signal noise, the algorithm consistently converges to the exact solution (see for instance Figs. 2-4). Neither extreme values of the parameters nor variations across different benchmark structures degrade performance (e.g., Figs. 4, 8, 9). This uniformity underscores that conjugate-gradient hypergradients remain stable throughout the optimization, yielding unbiased updates even when the problem is highly conditioned. In short, whenever the underlying model has a unique optimum, the solver reliably recovers it.

The nonlinear signal environments, in contrast, exhibit different behavior. Here the optimization succeeds in recovering the utilities: the agent and principal payoffs converge to the grid-search reference across a wide range of parameters (e.g., Figs. 37, 39). However, the corresponding contract parameters show a different trajectory. Their distances decrease initially but oftentimes plateau at non-negligible values, indicating that the learned contract does not always coincide with the specific reference solution (e.g., Figs. 28, 41). This divergence is not a numerical artifact but a reflection of the underlying setting.

Nonlinear signal models of the FOA type are not uniquely identifiable. Multiple contracts can implement the same distribution of outcomes, or at least equally good outcomes, and hence deliver identical utilities to the principal and agent. The plateauing of parameter distances we observe is the empirical manifestation of this non-identifiability. The algorithm converges to one of many payoff-equivalent optima, preserving the utilities but not the contract parameters themselves. Thus, in nonlinear environments, the method should be judged primarily on whether it recovers the correct utilities rather than the precise contract form.

Another subtlety in the nonlinear case is that convergence in utilities is less uniform. For some parameter regimes, the utility gaps shrink smoothly to near zero (e.g., Figs. 37, 39), while for others they remain noisy and hover at more moderate error levels (e.g., Figs. 28, 30). We attribute this to a combination of evaluation variance and sensitivity to the reference solution. Nonetheless, the trend is consistent across settings: the learned contracts deliver utilities close to optimal even when the parameter recovery is imperfect. This reinforces the view that the algorithm captures the economically relevant objects—payoffs—despite the absence of parameter identifiability.

Overall, the experiments show a dichotomy: on linear CARA–Normal benchmarks we exactly recover utilities and parameters; on nonlinear signals we are utility-consistent—matching optimal payoffs even when parameters aren't uniquely identified. Thus, the method gives exact recovery with unique solutions and payoff consistency when multiple contracts implement the outcome.

**Conclusions.** We introduced a scalable solver for principal–agent contract design by formulating hidden-action problems as bilevel max–max programs and applying implicit differentiation with conjugate gradients. The method avoids forming Hessians, recovers linear–quadratic optima to high precision, and extends to nonlinear and higher-dimensional settings where closed forms are unavailable. In nonlinear environments, it matches the economically relevant objects — the principal and agent utilities — even when several payoff-equivalent contracts exist. This establishes a practical bridge between classic contract theory and modern differentiable optimization.

**Limitations and scope.** Our analysis assumes smooth, correctly specified primitives for utilities, outcome laws, and constraints; non-smooth penalties or misspecification can challenge the regularity behind implicit differentiation. We study a locally concave inner problem (negative-definite Hessian block); flat or non-unique best responses may ill-condition the linear system, though damping and CG tolerances help. The present scope is static and single-agent; extending to dynamic, repeated, and multi-agent/competing-principal settings is a natural next step. Institutional frictions (limited liability, participation/renegotiation, budget balance, enforcement) can be added via tailored projections or penalties, with possible effects on convergence. In nonlinear signal models, utilities are identifiable but contract parameters need not be, so we target payoff consistency rather than exact recovery. Finally, estimates use sample-average approximation with common random numbers; hypergradients carry Monte Carlo noise but are stable with sensible batch sizes, damping, and CG tolerances. Finally, our grid-search references are reliable in low dimension yet motivate more scalable baselines.

## 5 REPRODUCIBILITY STATEMENT

We have aimed to make all components of our results easy to reproduce. The algorithmic details—including the sample-average approximation, implicit differentiation, CG solver, and common-random-numbers variance reduction—are specified in Sec. 3 and Algorithms 1–5, with all hyperparameters (learning rates/step sizes, iteration budgets, tolerances, CG damping, projections) reported in Sec. 4. Closed-form benchmarks (e.g., Holmström–Milgrom and extensions) and their derivations appear in App. B.1, which also describes the nonlinear environments and our grid-search ground-truth procedure (contract/action grids, boxes, and Monte Carlo/QMC settings). For stochastic estimation, we fix Sobol–QMC seeds and reuse common random numbers across all evaluations, as documented in Secs. 3.1–3.2. We provide, as anonymized supplementary material, a code repository containing: (i) reference implementations of Algs. 1–5 (matrix-free HVP/CG), (ii) environment generators and evaluators (CARA–Normal and nonlinear), (iii) exact-solution checkers for the closed-form cases, (iv) experiment configurations are documented in the captions of each experiment, and (v) plotting scripts to recreate the figures. We provide a readme file that describes how to reproduce some of the results in the paper. Assumptions for all theoretical claims are stated inline with each result, and complete proofs/derivations are included in the main text.

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

## A  LLM USAGE STATEMENT

Large Language Models (LLMs) were used solely as an assistive tool for improving the clarity and presentation of the manuscript (e.g., editing grammar, refining phrasing). All technical content, including theoretical derivations, proofs, experimental design, and analysis, was developed entirely by the authors. No parts of the paper were written or ideated by an LLM in a way that would constitute substantive scientific contribution, and no LLM was used to generate or fabricate results.

## B  ADDITIONAL RESULTS

To expand on the main-text experiments, we conducted a wide range of experiments with the settings outlined below.

### B.1  SETTINGS

We study hidden–action principal–agent models in which a risk–neutral principal offers a (possibly nonlinear) contract to a risk–averse agent who chooses an unobservable action $a$. Outcomes are noisy, compensation depends on observables, and the agent incurs a quadratic effort cost. We group environments into *linear* settings with closed-form optima and *nonlinear* settings where we compute reference solutions numerically.

**Ground-truth estimation and search domains for nonlinear contract-design settings.** When closed forms are unavailable, we approximate ground truth via a *nested grid search*: for each contract parameters $t$ on a rectangular grid, we estimate the agent's best response $a^\star(t)$ on a 1D action grid and then pick the $t$ that maximizes the principal's objective at $a^\star(t)$. Unless stated otherwise, we use common random numbers (CRN) within each search to reduce variance.

We reuse the following generic boxes (chosen to enforce wage floors for nonlinear utilities and to cover essentially all mass of $X$ given the noise scale):

$$\lambda \in [\, w_{\min},\, w_{\min}+8\,], \qquad \mu \in [\, 0,\, 8\,], \qquad a \in [\, a_0-6s,\, a_0+6s\,].$$

The contract grid is $100 \times 100$; the action grid has 200 points. When a different box is used (e.g., CRRA), we state the change explicitly below.

#### B.1.1  LINEAR SETTINGS

**Holmström–Milgrom.** Output $y = a + \varepsilon$ with $\varepsilon \sim \mathcal{N}(0, \sigma^2)$ and linear pay $w = s + by$. With CARA agent (risk aversion $r$) and effort cost $\frac{c}{2}a^2$,

$$u_1(a,t) = a - \tfrac{1}{2}rb^2\sigma^2 - \tfrac{1}{2}ca^2,$$
$$u_2(a,t) = s + ba - \tfrac{1}{2}rb^2\sigma^2 - \tfrac{1}{2}ca^2.$$

Imposing participation $u_2(a^\star, t) = U_{\text{res}}$ yields

$$b^\star = \frac{1}{1 + rc\sigma^2}, \quad a^\star = \frac{b^\star}{c}, \quad s^\star = U_{\text{res}} - \left[b^\star a^\star - \tfrac{1}{2}r(b^\star)^2\sigma^2 - \tfrac{1}{2}c(a^\star)^2\right].$$

**Insurance with prevention (self–protection).** Loss $\tilde{L} = (\ell - a) + \varepsilon$, $\varepsilon \sim \mathcal{N}(0, \sigma^2)$, linear indemnity $b\tilde{L}$, premium $s$.

$$u_1(a,t) = -(\ell - a) - \tfrac{1}{2}r(1-b)^2\sigma^2 - \tfrac{1}{2}ca^2,$$
$$u_2(a,t) = -(1-b)(\ell - a) - s - \tfrac{1}{2}r(1-b)^2\sigma^2 - \tfrac{1}{2}ca^2.$$

Closed-form optimum:

$$b^\star = \frac{rc\sigma^2}{1 + rc\sigma^2}, \quad a^\star = \frac{1}{c(1 + rc\sigma^2)}, \quad s^\star = -U_{\text{res}} - (1-b^\star)(\ell - a^\star) - \tfrac{1}{2}r(1-b^\star)^2\sigma^2 - \tfrac{1}{2}c(a^\star)^2.$$

**Imperfect performance measurement.** Signal $z = \alpha a + \varepsilon$, $\varepsilon \sim \mathcal{N}(0, \sigma^2)$; contract $w = s + bz$.

$$u_1(a, t) = va - \tfrac{1}{2}rb^2\sigma^2 - \tfrac{1}{2}ca^2, \quad u_2(a, t) = s + b(\alpha a) - \tfrac{1}{2}rb^2\sigma^2 - \tfrac{1}{2}ca^2.$$

Closed-form optimum:

$$b^\star = \frac{v\alpha}{v\alpha^2 + rc\sigma^2}, \quad a^\star = \frac{\alpha b^\star}{c}, \quad s^\star = U_{\text{res}} - \left[b^\star\alpha a^\star - \tfrac{1}{2}r(b^\star)^2\sigma^2 - \tfrac{1}{2}c(a^\star)^2\right].$$

**Relative performance (peer benchmark).** Two agents with common shock $\eta \sim \mathcal{N}(0, \tau^2)$ and idiosyncratic noise. Contract $w_i = s + b\, y_i + d\, y_j$. Let $\sigma_{\text{eff}}^2 = \frac{\sigma^2(\sigma^2 + 2\tau^2)}{\sigma^2 + \tau^2}$. Then

$$d^\star = -b^\star\frac{\tau^2}{\sigma^2 + \tau^2}, \quad b^\star = \frac{v}{v + rc\sigma_{\text{eff}}^2}, \quad a^\star = \frac{b^\star}{c},$$

and $s^\star$ from participation.

**Separable multitask.** $K$ tasks with signals $y_i = a_i + \varepsilon_i$, $\varepsilon_i \sim \mathcal{N}(0, \sigma_i^2)$. Contract $w = s + \sum_{i=1}^{K} b_i y_i$. Then, task-wise,

$$b_i^\star = \frac{v_i}{v_i + rc_i\sigma_i^2}, \qquad a_i^\star = \frac{b_i^\star}{c_i},$$

and $s^\star$ is pinned down by participation:

$$s^\star = U_{\text{res}} - \left[\sum_{i=1}^{K} b_i^\star a_i^\star - \tfrac{1}{2}r\sum_{i=1}^{K} b_i^{\star 2}\sigma_i^2 - \tfrac{1}{2}\sum_{i=1}^{K} c_i a_i^{\star 2}\right].$$

**Two signals.** Output is observed through two independent noisy signals with variances $\sigma_1^2, \sigma_2^2$; the contract is $w = s + b_1 y_1 + b_2 y_2$. Let the effective variance be the harmonic-mean aggregate

$$\sigma_{\text{eff}}^2 = \left(\sigma_1^{-2} + \sigma_2^{-2}\right)^{-1}.$$

Define $\beta^\star = \frac{v}{v + rc\sigma_{\text{eff}}^2}$. The optimal slopes split $\beta^\star$ in proportion to signal precisions:

$$b_1^\star = \beta^\star\frac{\sigma_1^{-2}}{\sigma_1^{-2} + \sigma_2^{-2}}, \qquad b_2^\star = \beta^\star\frac{\sigma_2^{-2}}{\sigma_1^{-2} + \sigma_2^{-2}},$$

and the induced action is $a^\star = \beta^\star/c$. The transfer $s^\star$ is pinned down by the participation constraint.

### B.1.2 NONLINEAR SETTINGS

**Logistic signal.** We use $w(x) = \lambda + \mu\,\sigma\left(\frac{x - a_0}{s}\right)$ with $\sigma(u) = \frac{1}{1 + e^{-u}}$ and floor $w_{\min} > 0$. Output takes the form $X = a + sZ$ (distribution varies by setting). The principal's objective is $u_1(a, t) = \mathbb{E}[X - w(X)]$; the agent's utility $u_2(a, t)$ varies below.

**Logistic signal with square-root wage utility.** $Z \sim \text{Logistic}(0, 1)$, $u_2(a, t) = \mathbb{E}[\sqrt{w(X)}] - \tfrac{1}{2}ca^2$. Ground truth via the generic boxes above; CRN with a single Logistic batch.

**Logistic signal with CRRA wage utility.** $Z \sim \text{Logistic}(0, 1)$, $u_2(a, t) = \mathbb{E}\left[\frac{w^{1-\gamma}}{1-\gamma}\right] - \tfrac{1}{2}ca^2$ (log $w$ in the $\gamma \to 1$ limit). Here we tighten the contract box to keep $w$ well within a stable range:

$$\lambda \in [\,w_{\min}, 3.0\,], \qquad \mu \in [\,0.20, 3.0\,], \qquad a \in [\,a_0 - 6s, a_0 + 6s\,].$$

We keep the same grid resolutions and CRN scheme.

**Laplace signal with thresholded wage utility.** $Z \sim \text{Laplace}(0, 1)$, $u_2(a, t) = \mathbb{E}[\sigma(\rho\,(w(X) - \theta))] - \tfrac{1}{2}ca^2$ with curvature $\rho > 0$ and reference $\theta$. We use the generic boxes; the $\pm 6s$ action range covers $> 99.7\%$ of the mass under Laplace noise. CRN with a single Laplace batch.

**Poisson signal.** Counts with mean $m = \exp(a)$ are approximated as $X \approx m + \sqrt{m}\,Z$, $Z \sim \mathcal{N}(0, 1)$. Agent is CARA over wages: $u_2(a, t) = \mathbb{E}[-\exp(-\rho w(X))] - \tfrac{1}{2}ca^2$. We use the generic contract box and center the action window at $a_0$ with width $\pm 6$ (independent of $s$ here due to the reparameterization). CRN with a single Normal batch.

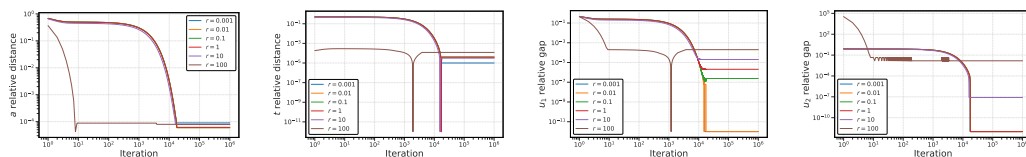

Figure 5: **Results for the Holmström–Milgrom model with varying $r$.** Fixed parameters: $c = 1.0$, $\sigma = 0.1$; varied parameter: $r \in \{10^i \mid i = -3, \ldots, 2\}$.

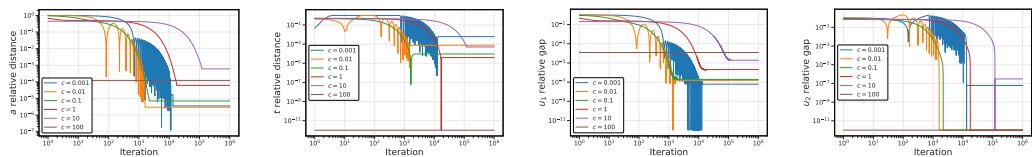

Figure 6: **Results for the Holmström–Milgrom model with varying $c$.** Fixed parameters: $r = 1.0$, $\sigma = 0.1$; varied parameter: $c \in \{10^i \mid i = -3, \ldots, 2\}$.

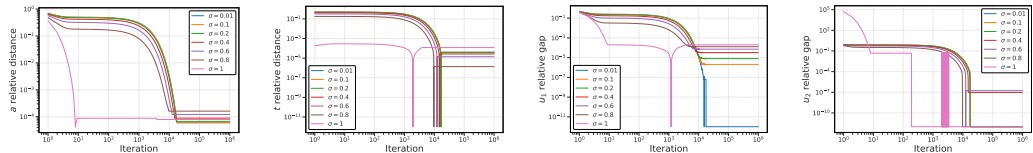

Figure 7: **Results for the Holmström–Milgrom model with varying $\sigma$.** Fixed parameters: $r = 1.0$, $c = 1.0$; varied parameter: $\sigma \in \{0.01, 0.1, 0.2, 0.4, 0.6, 0.8, 1\}$.

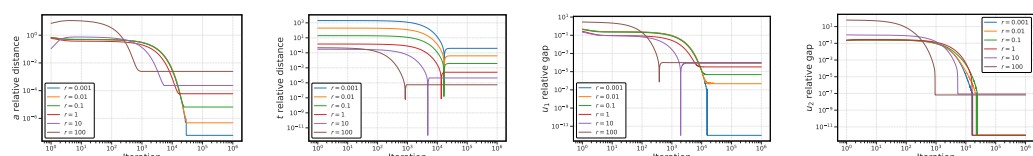

Figure 8: **Results for the insurance with prevention model with varying $r$.** Fixed parameters: $r = 1.0, c = 1.0, \sigma = 1.0, \ell = 1.0$; varied parameter: $r \in \{10^i \mid i = -3, \ldots, 2\}$.

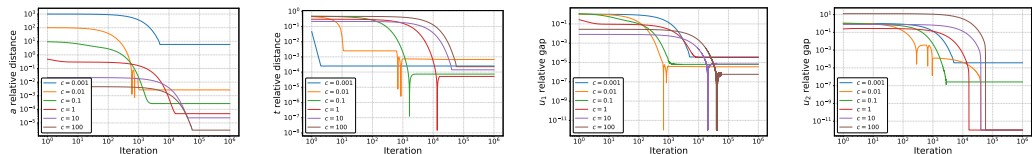

Figure 9: **Results for the insurance with prevention model with varying $c$.** Fixed parameters: $r = 1.0, c = 1.0, \sigma = 1.0, \ell = 1.0, U_{\mathrm{res}} = 0.0$; varied parameter: $c \in \{10^i \mid i = -3, \ldots, 2\}$.

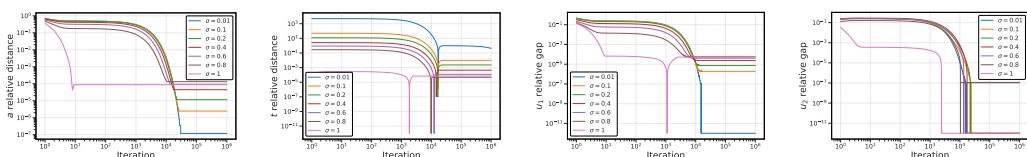

Figure 10: **Results for the insurance with prevention model with varying $\sigma$.** Fixed parameters: $r = 1.0, c = 1.0, \sigma = 1.0, \ell = 1.0, U_{\mathrm{res}} = 0.0$; varied parameter: $\sigma \in \{0.01, 0.1, 0.2, 0.4, 0.6, 0.8, 1\}$.

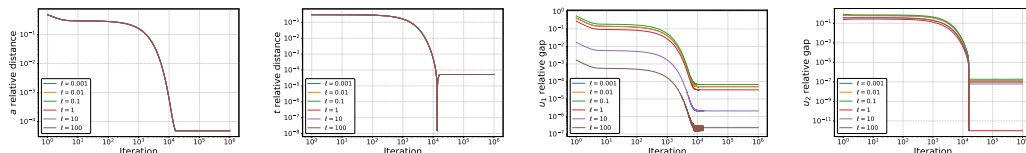

Figure 11: **Results for the insurance with prevention model with varying $\ell$.** Fixed parameters: $r = 1.0$, $c = 1.0$, $\sigma = 1.0$, $\ell = 1.0$, $U_{\mathrm{res}} = 0.0$; varied parameter: $\ell \in \{0.001, 0.01, 0.1, 1, 10, 100\}$.

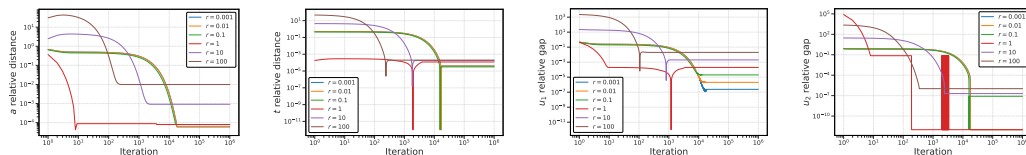

Figure 12: **Results for the imperfect performance measurement (CARA–Normal) model with varying $r$.** Fixed parameters: $r = 1.0$, $c = 1.0$, $\sigma = 1.0$, $\alpha = 1.0$, $v = 1.0$; varied parameter: $r \in \{10^i \mid i = -3, \ldots, 2\}$.

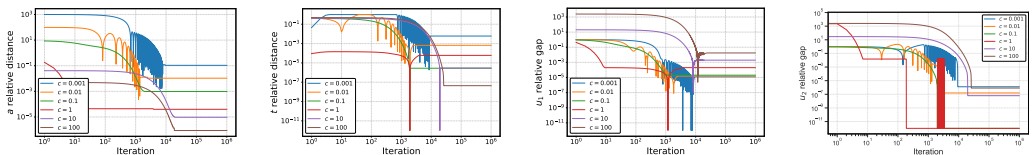

Figure 13: **Results for the imperfect performance measurement (CARA–Normal) model with varying $c$.** Fixed parameters: $r = 1.0$, $c = 1.0$, $\sigma = 1.0$, $\alpha = 1.0$, $v = 1.0$; varied parameter: $c \in \{10^i \mid i = -3, \ldots, 2\}$.

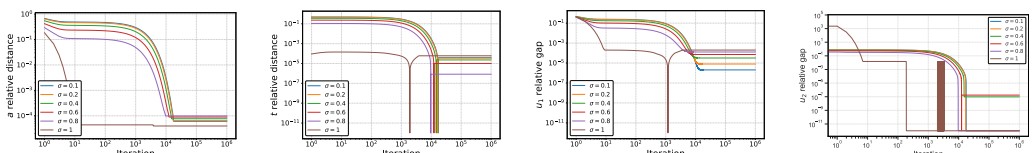

Figure 14: **Results for the imperfect performance measurement (CARA–Normal) model with varying $\sigma$.** Fixed parameters: $r = 1.0$, $c = 1.0$, $\sigma = 1.0$, $\alpha = 1.0$, $v = 1.0$; varied parameter: $\sigma \in \{0.01, 0.1, 0.2, 0.4, 0.6, 0.8, 1.0\}$.

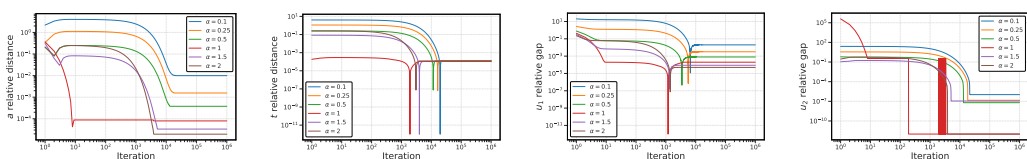

Figure 15: **Results for the imperfect performance measurement (CARA–Normal) model with varying $\alpha$.** Fixed parameters: $r = 1.0$, $c = 1.0$, $\sigma = 1.0$, $\alpha = 1.0$, $v = 1.0$; varied parameter: $\alpha \in \{0.1, 0.25, 0.5, 1, 1.5, 2\}$.

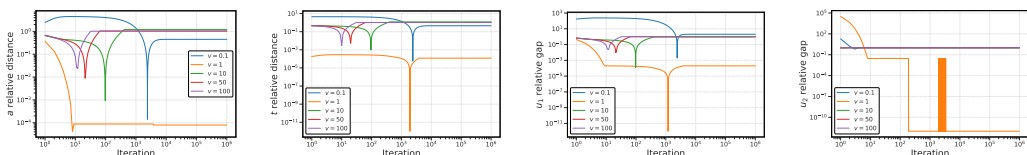

Figure 16: **Results for the imperfect performance measurement (CARA–Normal) model with varying $v$.** Fixed parameters: $r = 1.0$, $c = 1.0$, $\sigma = 1.0$, $\alpha = 1.0$, $v = 1.0$; varied parameter: $v \in \{0.1, 1, 10, 50, 100\}$.

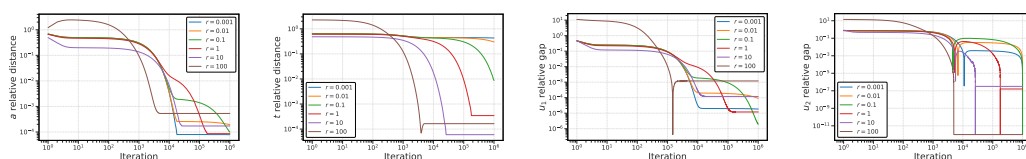

Figure 17: **Results for the relative performance model with varying $r$.** Fixed parameters: $c = 1.0$, $\sigma = 0.2$, $\tau = 0.2$, $v = 1.0$, $a_{\text{peer}} = 0.1$; varied parameter: $r \in \{10^i \mid i = -3, \ldots, 2\}$.

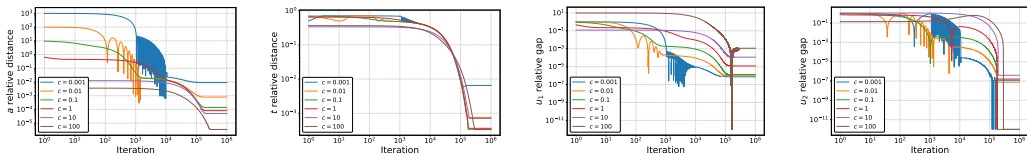

Figure 18: **Results for the relative performance model with varying $c$.** Fixed parameters: $r = 1.0$, $\sigma = 0.2$, $\tau = 0.2$, $v = 1.0$, $a_{\text{peer}} = 0.1$; varied parameter: $c \in \{10^i \mid i = -3, \ldots, 2\}$.

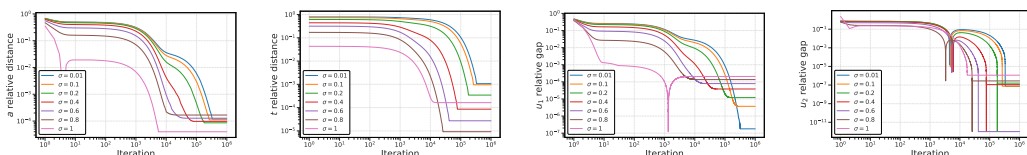

Figure 19: **Results for the relative performance model with varying $\sigma$.** Fixed parameters: $r = 1.0$, $c = 1.0$, $\tau = 0.2$, $v = 1.0$, $a_{\text{peer}} = 0.1$; varied parameter: $\sigma \in \{0.01, 0.1, 0.2, 0.4, 0.6, 0.8, 1\}$.

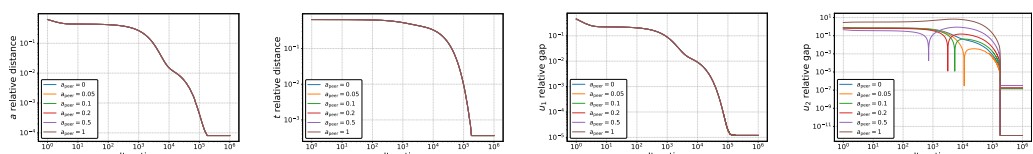

Figure 20: **Results for the relative performance model with varying $a_{\text{peer}}$.** Fixed parameters: $r = 1.0$, $c = 1.0$, $\sigma = 0.2$, $\tau = 0.2$, $v = 1.0$; varied parameter: $a_{\text{peer}} \in \{0.0, 0.05, 0.1, 0.2, 0.5, 1.0\}$.

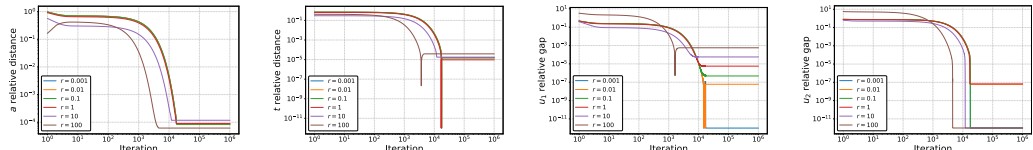

Figure 21: **Results for the separable multitask model with varying $r$.** Fixed parameters: $c = 1.0$, $\sigma = 0.2$, $v = 1.0$; varied parameter: $r \in \{10^i \mid i = -3, \ldots, 2\}$.

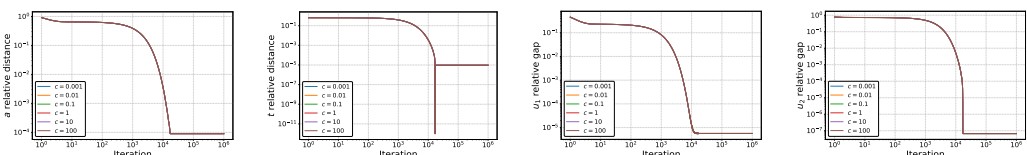

Figure 22: **Results for the separable multitask model with varying $c$.** Fixed parameters: $r = 1.0$, $\sigma = 0.2$, $v = 1.0$; varied parameter: $c \in \{10^i \mid i = -3, \ldots, 2\}$.

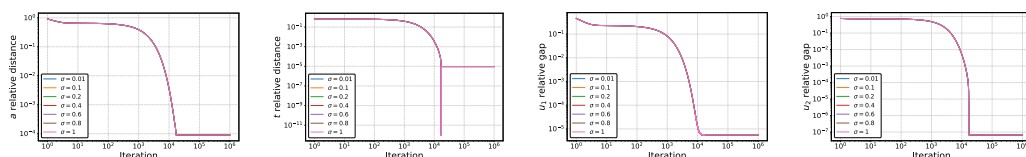

Figure 23: **Results for the separable multitask model with varying $\sigma$.** Fixed parameters: $r = 1.0$, $c = 1.0$, $v = 1.0$; varied parameter: $\sigma \in \{0.01, 0.1, 0.2, 0.4, 0.6, 0.8, 1\}$.

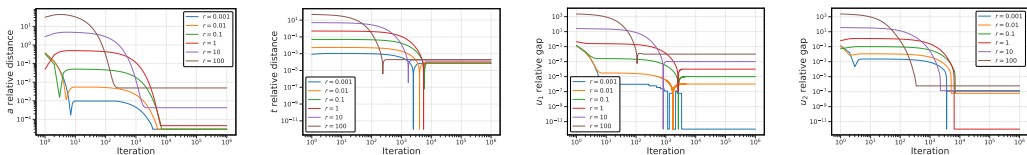

Figure 24: **Results for the two signals model with varying $r$.** Fixed parameters: $c = 1.0$, $\sigma_1 = 1.0$, $\sigma_2 = 1.0$, $U_{res} = 0.0$; varied parameter: $r \in \{10^i \mid i = -3, \dots, 2\}$.

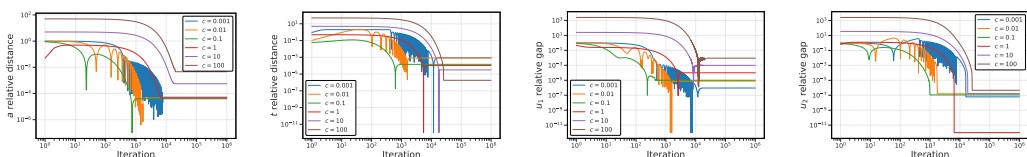

Figure 25: **Results for the two signals model with varying $c$.** Fixed parameters: $r = 1.0$, $\sigma_1 = 1.0$, $\sigma_2 = 1.0$, $U_{res} = 0.0$; varied parameter: $c \in \{10^i \mid i = -3, \dots, 2\}$.

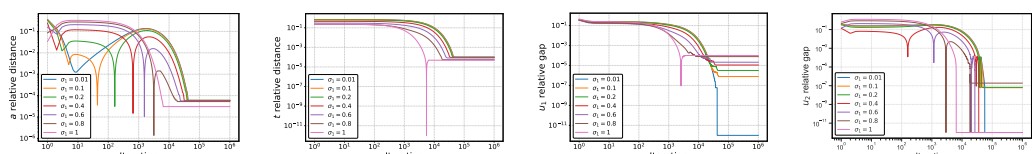

Figure 26: **Results for the two signals model with varying $\sigma_1$.** Fixed parameters: $r = 1.0$, $c = 1.0$, $\sigma_2 = 1.0$, $U_{res} = 0.0$; varied parameter: $\sigma_1 \in \{0.01, 0.1, 0.2, 0.4, 0.6, 0.8, 1.0\}$.

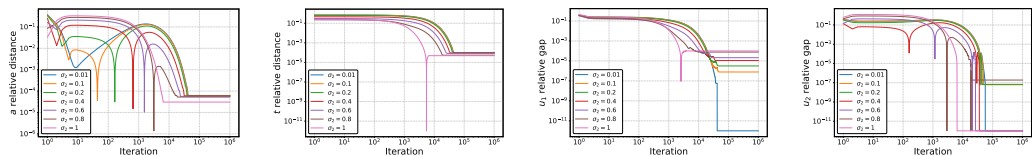

Figure 27: **Results for the two signals model with varying $\sigma_2$.** Fixed parameters: $r = 1.0$, $c = 1.0$, $\sigma_1 = 1.0$, $U_{res} = 0.0$; varied parameter: $\sigma_2 \in \{0.01, 0.1, 0.2, 0.4, 0.6, 0.8, 1.0\}$.

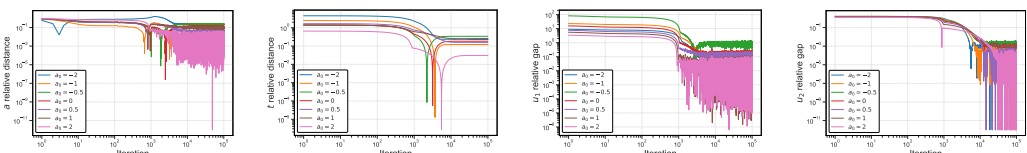

Figure 28: **Results for the logistic signal model with varying $a_0$.** Fixed parameters: $c = 0.25$, $s = 1.0$, $w_{\min} = 0.25$; varied parameter: $a_0 \in \{0.01, 0.1, 0.2, 0.4, 0.6, 0.8, 1\}$.

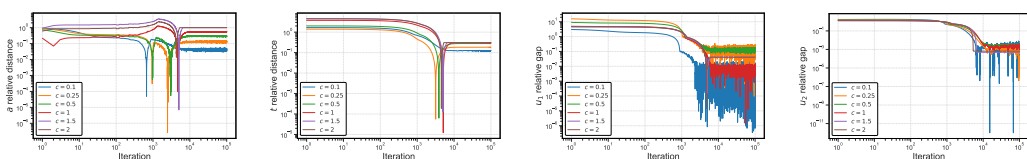

Figure 29: **Results for the logistic signal model with varying $c$.** Fixed parameters: $a_0 = 0.0$, $s = 1.0$, $w_{\min} = 0.25$; varied parameter: $c \in \{0.1, 0.2, 0.3, 0.5, 0.8, 1\}$.

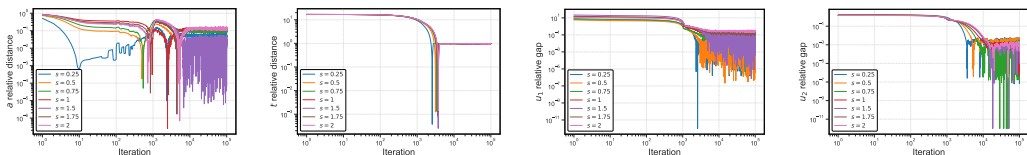

Figure 30: **Results for the logistic signal model with varying $s$.** Fixed parameters: $a_0 = 0.0$, $c = 0.25$, $w_{\min} = 0.25$; varied parameter: $s \in \{0.5, 0.75, 1, 1.25, 1.5, 2\}$.

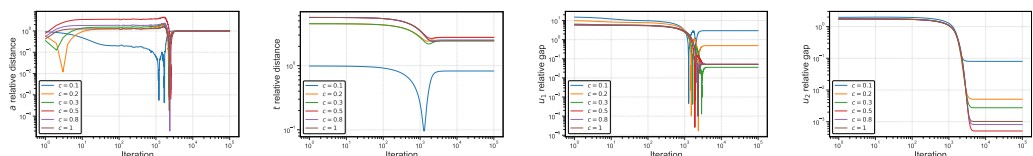

Figure 31: **Results for the logistic signal with square-root wage utility model with varying $c$.** Fixed parameters: $s = 1.0$, $w_{\min} = 0.2$, $a_0 = 0.0$; varied parameter: $c \in \{0.1, 0.2, 0.3, 0.5, 0.8, 1\}$.

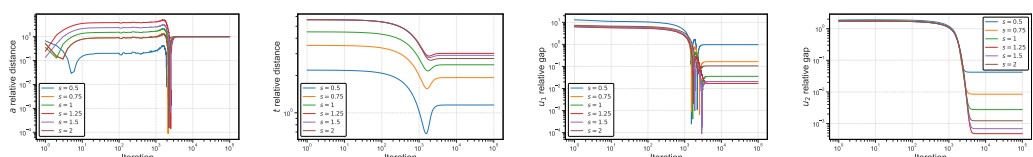

Figure 32: **Results for the logistic signal with square-root wage utility model with varying $s$.** Fixed parameters: $c = 0.3$, $w_{\min} = 0.2$, $a_0 = 0.0$; varied parameter: $s \in \{0.5, 0.75, 1, 1.25, 1.5, 2\}$.

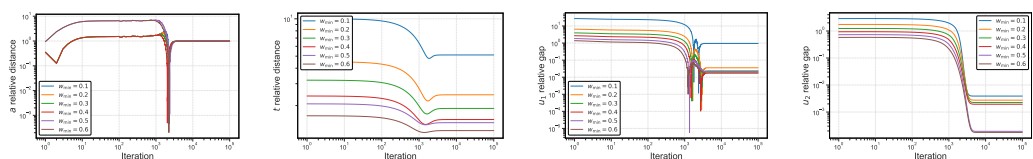

Figure 33: **Results for the logistic signal with square-root wage utility model with varying $w_{\min}$.** Fixed parameters: $c = 0.3$, $s = 1.0$, $a_0 = 0.0$; varied parameter: $w_{\min} \in \{0.1, 0.15, 0.2, 0.3, 0.4\}$.

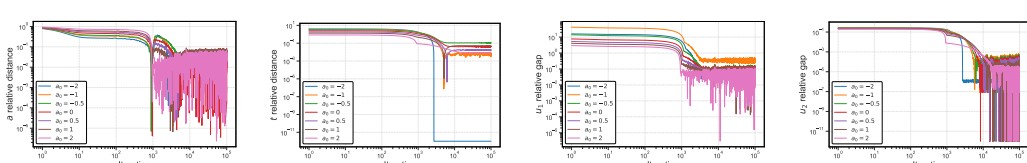

Figure 34: **Results for the logistic signal with CRRA wage utility model with varying $a_0$.** Fixed parameters: $s = 1.0$, $c = 0.3$, $w_{\min} = 0.2$, $\gamma = 1.20$; varied parameter: $a_0 \in \{0.01, 0.1, 0.2, 0.4, 0.6, 0.8, 1\}$.

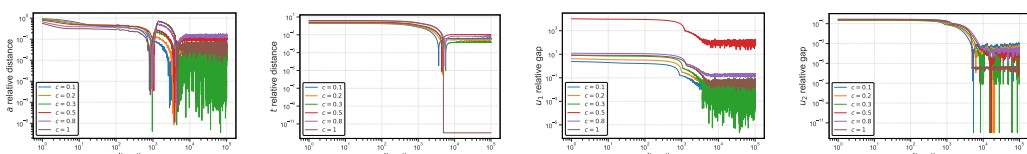

Figure 35: **Results for the logistic signal with CRRA wage utility model with varying $c$.** Fixed parameters: $s = 1.0$, $a_0 = 0.0$, $w_{\min} = 0.2$, $\gamma = 1.20$; varied parameter: $c \in \{0.1, 0.2, 0.3, 0.5, 0.8, 1\}$.

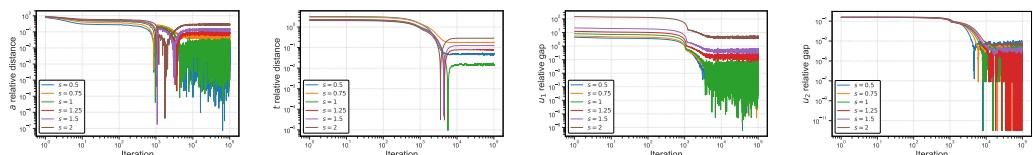

Figure 36: **Results for the logistic signal with CRRA wage utility model with varying $s$.** Fixed parameters: $c = 0.3$, $a_0 = 0.0$, $w_{\min} = 0.2$, $\gamma = 1.20$; varied parameter: $s \in \{0.5, 0.75, 1, 1.25, 1.5, 2\}$.

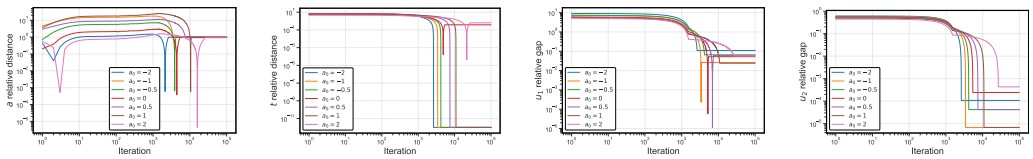

Figure 37: **Results for the Laplace signal with threshold wage utility model with varying $a_0$.** Fixed parameters: $s = 1.0$, $c = 0.3$, $w_{\min} = 0.2$, $\rho = 1.25$, $\theta = 0.0$; varied parameter: $a_0 \in \{0.01, 0.1, 0.2, 0.4, 0.6, 0.8, 1\}$.

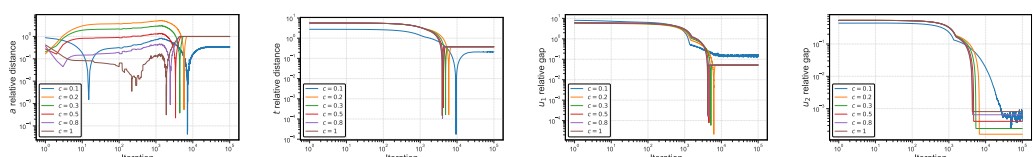

Figure 38: **Results for the Laplace signal with threshold wage utility model with varying $c$.** Fixed parameters: $s = 1.0$, $a_0 = 0.0$, $w_{\min} = 0.2$, $\rho = 1.25$, $\theta = 0.0$; varied parameter: $c \in \{0.1, 0.2, 0.3, 0.5, 0.8, 1\}$.

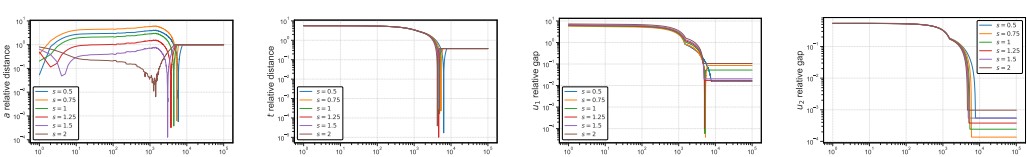

Figure 39: **Results for the Laplace signal with threshold wage utility model with varying $s$.** Fixed parameters: $c = 0.3$, $a_0 = 0.0$, $w_{\min} = 0.2$, $\rho = 1.25$, $\theta = 0.0$; varied parameter: $s \in \{0.5, 0.75, 1, 1.25, 1.5, 2\}$.

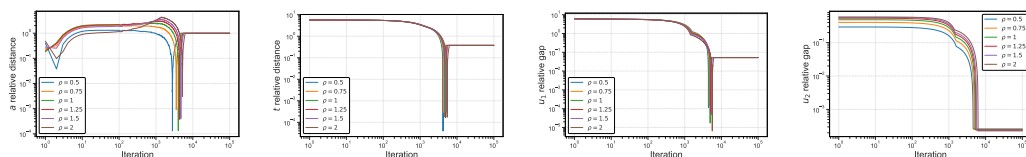

Figure 40: **Results for the Laplace signal with threshold wage utility model with varying** $\rho$**.** Fixed parameters: $s = 1.0$, $c = 0.3$, $a_0 = 0.0$, $w_{\min} = 0.2$, $\theta = 0.0$; varied parameter: $\rho \in \{0.5, 0.75, 1, 1.25, 1.5, 2\}$.

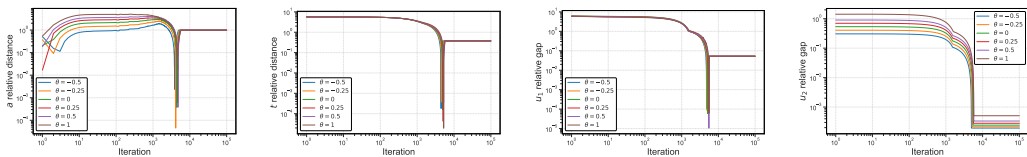

Figure 41: **Results for the Laplace signal with threshold wage utility model with varying** $\theta$**.** Fixed parameters: $s = 1.0$, $c = 0.3$, $a_0 = 0.0$, $w_{\min} = 0.2$, $\rho = 1.25$; varied parameter: $\theta \in \{-0.5, -0.25, 0, 0.25, 0.5, 1\}$.

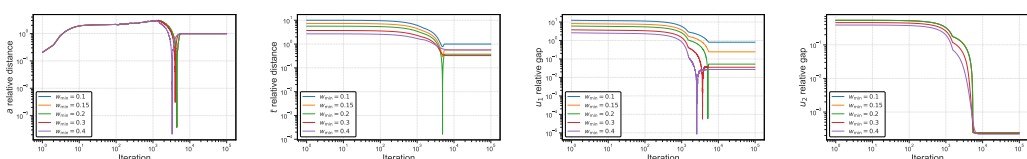

Figure 42: **Results for the Laplace signal with threshold wage utility model with varying** $w_{\min}$**.** Fixed parameters: $s = 1.0$, $c = 0.3$, $a_0 = 0.0$, $\rho = 1.25$, $\theta = 0.0$; varied parameter: $w_{\min} \in \{0.1, 0.15, 0.2, 0.3, 0.4\}$.

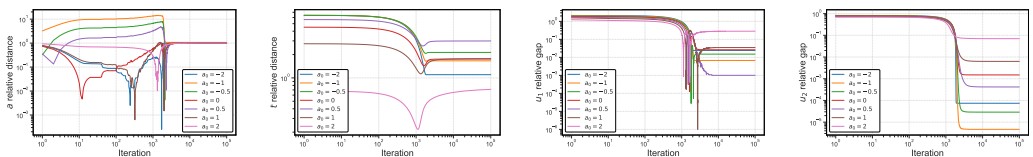

Figure 43: **Results for the Poisson signal (Mean–Exp parameterization) with varying** $a_0$**.** Fixed parameters: $c = 0.30$, $w_{\min} = 0.20$, $\rho = 1.00$; varied parameter: $a_0 \in \{-2, -1, -0.5, 0, 0.5, 1, 2\}$.

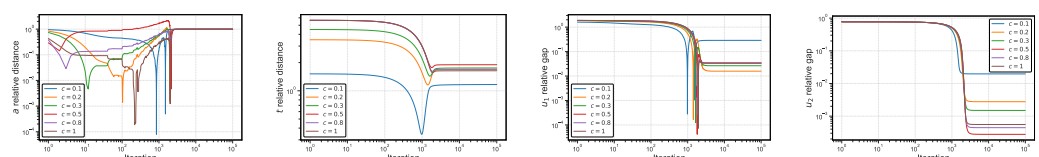

Figure 44: **Results for the Poisson signal (Mean–Exp parameterization) with varying** $c$**.** Fixed parameters: $a_0 = 0.0$, $w_{\min} = 0.20$, $\rho = 1.00$; varied parameter: $c \in \{0.1, 0.2, 0.3, 0.5, 0.8\}$.

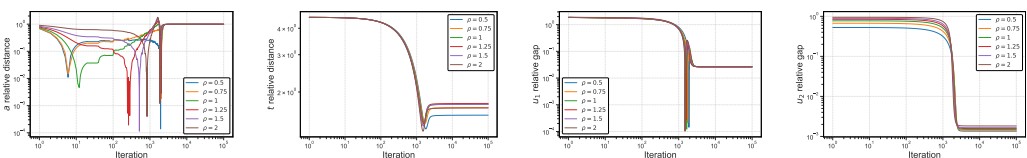

Figure 45: **Results for the Poisson signal (Mean–Exp parameterization) with varying** $\rho$**.** Fixed parameters: $c = 0.30$, $a_0 = 0.0$, $w_{\min} = 0.20$; varied parameter: $\rho \in \{0.5, 0.75, 1, 1.25, 1.5, 2\}$.

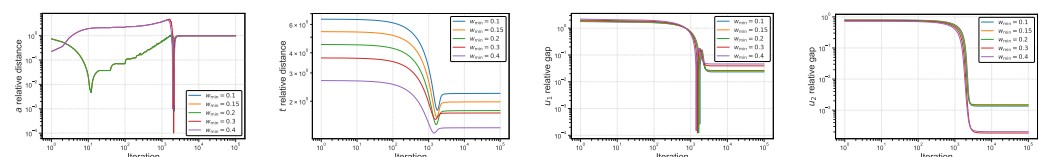

Figure 46: **Results for the Poisson signal (Mean–Exp parameterization) with varying** $w_{\min}$**.** Fixed parameters: $c = 0.30$, $a_0 = 0.0$, $\rho = 1.00$; varied parameter: $w_{\min} \in \{0.1, 0.15, 0.2, 0.3, 0.4\}$.

