# OpenReview forum: "Scalable Principal–Agent Contract Design via Gradient-Based Optimization"
_ICLR.cc/2026/Conference — ICLR 2026 Conference Withdrawn Submission_

### Official Review · Reviewer_m46r · 2025-10-23

**Soundness:** 3
**Presentation:** 1
**Contribution:** 2
**Rating:** 2
**Confidence:** 3

**Summary:**

This paper proposes a bilevel optimization approach to estimate optimal contracts in principal-agent settings.
In particular, it proposes an implicit differentiation based algorithm, reminiscent of typical bilevel optimization methods, and run multiple experiments to illustrate the power of this method on numerous principal-agent benchmarks.

**Strengths:**

The bilevel formulation of the contract design in principal agent problem is natural and should probably to improved algorithms for computing such contracts with respect to existing literature. The paper also presents many experiments (in the appendix section), supporting their method.

**Weaknesses:**

I am not sure to understand the real contribution/scope of this paper. If I understood clearly, this paper proposes to adopt a bilevel approach to compute optimal contracts in principal-agent problems, and the authors then propose such a bilevel optimization algorithm.

However, I don't understand why the authors propose a "new" bilevel optimization method, instead of applying classical bilevel optimization algorithms and benefiting from a rich literature on the topic. In consequence, I am not sure if the main contribution of this work is a "new bilevel optimization method" (Section 3), in which case this paper belongs to the field of bilevel optimization and comparisons with existing methods is missing.

If on the other hand the main contribution of this paper is to directly apply existing bilevel optimization algorithms to principal-agent problems, then the experimental section 4 is clearly not emphasized enough. If the goal is to build solid experiments and benchmarks, the current figures are clearly not enough: there is no comparison with existing methods (either bilevel or existing contract algorithms), there is no clear visualisation of the many experiments run by the authors (I would for example suggest a table/histogram summarizing some average performance of the model). Moreover, the figure in the main text is only for Holmstrom-Milgrom linear-quadratic model, which is stated as an easy (and solved) problem by the authors.

**Questions:**

What is the main contribution of this work? Is it propose a new bilevel optimization algorithm or to apply bilevel methods for computing optimal contract?

---

### Official Review · Reviewer_HUsX · 2025-11-01

**Soundness:** 3
**Presentation:** 2
**Contribution:** 2
**Rating:** 4
**Confidence:** 2

**Summary:**

This paper presents a computational framework to solve principal-agent contract design problem.  Traditional contract design models admit closed-form solutions only under restrictive assumptions, such as linear environments and quadratic costs. To efficiently solve more general contract design problems (which do not have closed-form solutions), the paper uses bilevel optimization methods in the literature: in particular, Hessian-vector products (HVP) and conjugate gradient (CG), which alleviate the need to invert a Hessian matrix (which is expensive).  Experiments show that the proposed methods recover the optimal solution in the settings with closed-form solutions, and achieve optimal utilities (compared to grid search) in the settings without closed-form solutions where the optimal contract might be non-unique.

**Strengths:**

(S1) While the formulation of contract design as bilevel optimization is not a new idea, this paper applies modern bilevel optimization techniques (e.g., HVP and CG) to efficiently solve contract design bilevel optimization problems.  This method differs from previous approaches relying on surrogate learning and discretization, and seems to be better for large-scale problem with continuous action space of the agent.

**Weaknesses:**

(W1) **Lack of theoretical guarantee:**  The paper didn't provide theoretical proof that the proposed algorithms can converge to the optimal solution.  A basic question might be: assuming that the Hessian and gradient are estimated perfectly, can the algorithm always find an optimal solution in the basic linear environment where closed-form solutions exist?  Even this basic question is not answered.  Neither does the paper analyze the effect of the estimation error for gradients and Hessians on the algorithmic performance.  Although the algorithms find good solutions experimentally, the lack of theoretical convergence guarantee and robustness analysis might limit the practicality of the algorithms.

(W2) It is unclear to me what is special about contract design in the context of this paper.  This paper formulates contract design as a bilevel optimization problem: $\max_t u_1(a^\star(t), t)$ s.t. $a^\star(t) \in \arg\max_a u_2(a, t)$.  This general form captures any bilevel optimization problem.  The theoretical derivations are also for general bilevel problems.  The authors didn't seem to exploit any specific properties of the contract design problem.  **This approach might be too general to be useful in real-world contract design problems with specific structures.**

(W3) **Limited economic insight:**  Although the computational contribution is strong, the paper offers little new economic understanding. It primarily replicates known benchmarks rather than producing new theoretical insights about optimal contracts.  Since the algorithms can identify nearly optimal contract in the cases without closed-form solution, it might be worthwhile to illustrate the economic structure of such solutions.

**Questions:**

## Questions for the authors

I don't have specific questions, but I'm happy to see authors' responses to my criticisms.



### Suggestions

Typos:

* Line 185: "C(a, t)" is mentioned in the text but not in the displayed equation (1).
* Line 215: "CG solve" -> "CG solver"
* Some parts of the writing are not clean.  For example, "Conjugate Gradient" -- a key technical component -- is mentioned in many places but never defined mathematically.  I think it refers to the $v$ in Equation (9), but I am not sure because I am not familiar with bilevel optimization literature.  The "Common Random Numbers" technique is unnecessarily repeated in pages 5 and 6. "SPD fix and damping" is repeated in page 5 (260-264) and page 6.

---

### Official Review · Reviewer_BTat · 2025-11-01

**Soundness:** 2
**Presentation:** 3
**Contribution:** 1
**Rating:** 4
**Confidence:** 4

**Summary:**

The paper considers the min-max problem arising in contract design and instantiates a gradient-based algorithm to optimize it.
In particular, they rely on implicit differentiation and conjugate gradients. In the case of known analytical solutions, the problem efficiently recovers those solutions. The method is also used in settings where no known analytical solutions exist.

**Strengths:**

Learning problems in economic settings in a very relevant and timely topic. Moreover, the paper is very well written.

**Weaknesses:**

The biggest weakness is the technical novelty. It seems that the method designed by the authors does not really exploit any intrinsic feature of contract design and could be applied to any Stackelberg game. This is also evident from the question asked before section 3, which asks how to design efficient optimization algorithms to solve general bilevel max–max problems, not specifically for contract design.
While the empirical results are solid, there is no new theoretical analysis, e.g., no convergence guarantees or convergence analysis.

**Questions:**

* Are there other methods for bilevel min-max problems? Why can’t those be used?
* Does your method provably converge to known analytical solutions, or is it only empirical evidence?
* Are there any theoretical guarantees that your algorithm gets? Like convergence to local min-max points?

---

### Note · Authors · 2025-11-25

I have read and agree with the venue's withdrawal policy on behalf of myself and my co-authors.